# A New Strategy for Consumption of Functional Lipids from *Ericerus pela* (Chavannes): Study on Microcapsules and Effervescent Tablets Containing Insect Wax–Derived Policosanol

**DOI:** 10.3390/foods12193567

**Published:** 2023-09-26

**Authors:** Yiwen Liu, Hong Zhang, Juan Xu, Rui He, Jinju Ma, Chiqing Chen, Lanxiang Liu

**Affiliations:** 1Institute of Highland Forest Science, Chinese Academy of Forestry, Kunming 650233, China; liuyiwen90@outlook.com (Y.L.);; 2Research Center of Efficient Breeding and Deep Processing Engineering Technology of Gallnut, National Forestry and Grassland Administration, Wufeng 443400, China; 3Key Laboratory of Breeding and Utilization of Resource Insects, National Forestry and Grassland Administration, Kunming 650233, China; 4Research Center of Engineering and Technology of Characteristic Forest Resources, National Forestry and Grassland Administration, Kunming 650233, China

**Keywords:** edible insect, functional insect lipids, insect wax–derived policosanol, microcapsules, effervescent tablets

## Abstract

In this study, we addressed various challenges associated with the consumption of functional lipids from the *Ericerus pela* (Chavannes), including unfavorable taste, insolubility in water, difficulty in oral intake, low bioavailability, and low psychological acceptance. Our study focused on the microencapsulation of policosanol, the key active component of insect wax, which is a mixture of functional lipids secreted by the *Ericerus pela* (Chavannes). We developed two innovative policosanol products, microcapsules, and effervescent tablets, and optimized their preparation conditions. We successfully prepared microcapsules containing insect wax–derived policosanol using the spray-drying method. We achieved 92.09% microencapsulation efficiency and 61.67% powder yield under the following conditions: maltodextrin, starch sodium octenyl succinate, and (2-hydroxy)propyl-β-cyclodextrin (HPβCD) at a ratio of 1:1:1, core-to-wall materials at a ratio of 1:10, 15% solid content, spray dryer feed temperature at 60 °C, inlet air temperature at 140 °C, and hot-air flow rate at 0.5 m^3^/min. The microcapsules exhibited a regular spherical shape with a minimal water content (1.82%) and rapid dispersion in water (within 143.5 s). These microcapsules released policosanol rapidly in simulated stomach fluid. Moreover, effervescent tablets were prepared using the policosanol-containing microcapsules. The tablets showed low friability (0.32%), quick disintegration in water (within 99.5 s), and high bubble volume. The microcapsules and effervescent tablets developed in this study presented effective solutions to the insolubility of policosanol in water. These products were portable and offered customizable tastes to address the psychological discomfort related to insect-based foods, thus providing a novel strategy for the consumption and secondary processing of insect lipids.

## 1. Introduction

More than 1900 insect species with high nutritional value are consumed as food worldwide. As the global population increases, insects present a potential and important food source because of their sustainable production and limited environmental impact [1,2]. Edible insects are rich in proteins, amino acids, lipids, carbohydrates, mineral elements, and vitamins. These insects are consumed as health-promoting foods with the effects of immunity regulation, fatigue reduction, antimutagenesis, and antioxidation, as well as lowering blood lipid and glucose levels [2,3,4]. The significant nutritional value of many insects is attributed to their high content of lipids, along with high levels of naturally active fatty acids, phospholipids, and fat-soluble vitamins (vitamins A, D, and E). The remarkably diverse species of insects, known for their rapid reproduction, abundant lipid content, and balanced fatty acid composition, make insect lipids a highly promising source of high-quality edible oils and fats. Food products have been developed based on insect lipids, such as beeswax and silkworm pupa oil [5]. Moreover, insects can selectively absorb and store specific biologically active substances, paving the way for the development and efficient utilization of functional insect lipids. Consequently, interest is growing in the exploration and application of edible insect lipids worldwide [6,7,8].

*Ericerus pela* (Chavannes) is one of the earliest economically viable scale insects. The insect is widely distributed in China, ranging from subtropical to temperate zones, and is also found in Vietnam, Japan, and the Korean Peninsula [9]. *Ericerus pela* (Chavannes) have high nutritional value. Their eggs contain significant amounts of proteins (44.67%), lipids (24.85%), sugars (1.54%), cellulose (8.90%), lecithin (9.22%), and polysaccharides (7.00%), along with 17 amino acids (including seven essential amino acids) [10,11,12]. These nutrients offer various health-promoting effects, such as immune regulation, antimutation, and fatigue reduction [13,14]. In many parts of the world, *Ericerus pela* (Chavannes) have been traditionally consumed in the form of insect powder and tea. However, direct consumption has been hindered by psychological barriers [2]. Male *Ericerus pela* (Chavannes) is a well-known wax producer and secrete a considerable amount of insect wax during the second-instar nymphal stage. The insect wax surrounding the insect’s body is usually consumed along with the insects [12,15]. Insect wax is a mixture of lipids that are secreted and accumulated on the scales. Insect wax is glossy, impurity-free, and chemically stable, 93–95% of which is composed of wax esters formed by even carbon-numbered (C_26_–C_30_) saturated fatty acids and corresponding saturated fatty alcohols [16,17]. Insect wax has the effect of moistening the lungs; invigorating the intestines and stomach; killing parasites; alleviating hematoma, bleeding, and hematochezia; and reducing fatigue [16,18]. Insect wax is commonly consumed directly in regions where *Ericerus pela* (Chavannes) scales are prevalent. For example, insect wax is added to hotpot seasoning in the Sichuan and Yunnan provinces of China, while insect wax is used as an ingredient in cooking soup or rice in the Wuling Mountain area of China. Insect wax, however, has an unfavorable taste, negligible water solubility, low absorption rate, and limited psychological acceptance, which have hindered its widespread adoption. To address these issues, it is critical to improve insect wax palatability and bioavailability, while reducing the disgust and discomfort related to insect-based food. Policosanol is the primary active component in insect wax and consists mainly of hexacosanol and octacosanol [16,19]. Policosanol is formed by saponification or reduction of insect wax and has many important physiological activities [20,21,22]. Hexacosanol is known to induce neuronal maturation, protecting and regenerating nerves, slowing the degeneration of cholinergic neurons, and preventing neurodegenerative diseases (e.g., Alzheimer’s disease). Octacosanol reduces cholesterol and triglyceride levels, relieves fatigue, increases the basal metabolic rate, and enhances strength and endurance. In addition, the combination of hexacosanol and octacosanol has been shown to inhibit atherosclerosis and thrombosis [23,24,25]. Ma et al. confirmed that both insect wax and policosanol are safe for consumption as they did not exhibit acute oral toxicity or potential chromosomal teratogenicity [16]. Insect wax and policosanol, however, are insoluble in water and only slightly soluble in some toxic organic solvents, leading to low processability and absorption, difficulty in oral intake, and low bioavailability [19].

Consequently, in this study, we focused on the development of two novel products, microcapsules, and effervescent tablets, using the policosanol extracted from insect wax. We also determined their preparation conditions and quality indicators. We not only improved the taste, bioavailability, portability, and ease of intake of insect wax–derived policosanol but also reduced the discomfort associated with insect-based food (Figure 1). Our results provide new insight into the processing, formulation, and promotion of *Ericerus pela* (Chavannes) lipids and edible insects.

## 2. Materials and Methods

### 2.1. Chemicals and Reagents

Insect wax–derived policosanol was prepared from insect wax via reduction reaction using LiAlH_4_ according to the method previously published by our team [22], mainly consisting of hexacosanol (56.0%) and octacosanol (32.0%) [16,22]. Soybean oil (refined first grade, density of 0.917, food grade) was purchased from Yihai Kerry Arawana Holdings Co., Ltd. (Shanghai, China). Lecithin (from soybean, >98%), polyglycerin ricinolate, (2-hydroxy)propyl-β-cyclodextrin (HPβCD) (degree of substitution, 6.3), 1,2-propylene glycol, tween-20, maltodextrin, and sodium starch octenyl succinate were purchased from Shanghai Aladdin Biochemical Technology Co., Ltd. (Shanghai, China).

### 2.2. Preparation of Microencapsules Containing Insect Wax–Derived Policosanol

We completely dissolved 1.00 g of insect wax–derived policosanol in 2.00 g of soybean oil (85 °C) and mixed it with 2.25 g of polyglyceryl ricinoleate, 2.27 g of Tween-20, 1.50 g of lecithin, and 6.00 g of 1,2-propanediol. The resulting mixture was stirred and emulsified for 20 min and then added dropwise into an aqueous solution (85 °C) containing maltodextrin, starch sodium octenyl succinate, HPβCD, and 240 mL deionized water. The mixture was stirred and emulsified at 550 rpm for 4 h and 8000 rpm for 1 h to obtain a microemulsion, which was spray dried by a spray dryer (SD-1000, Tokyo Rikakikai Co., Ltd., Tokyo, Japan) to yield the microcapsules containing policosanol. We optimized the preparation conditions of the microcapsule.

#### 2.2.1. Wall Material Ratio and Core-to-Wall Material Ratio

We examined the microencapsulation efficiency and powder yield at the wall material ratios of 1:1:0, 1:0:1, 0:1:1, 1:1:1, 1:2:1, and 1:3:1 (m/m) of maltodextrin, starch sodium octenyl succinate, and HPβCD. At this time, the core wall ratio was 1:10, and the solid content was 15%. Thus, the optimal wall material ratio was obtained. Then, we determined microencapsulation efficiency and powder yield at the core-to-wall material ratios, namely policosanol-to-composite wall material ratios, of 1:3, 1:5, 1:7, 1:10, and 1:12 (m/m) to select the optimal core-to-wall material ratio.

#### 2.2.2. Solid Content and Feed Temperature

We determined microencapsulation efficiency and powder yield at a solid content of 5%, 10%, 15%, 20%, and 25% and feed temperatures of 40 °C, 50 °C, 60 °C, 70 °C, and 80 °C. We selected the optimal solid content and feed temperature.

#### 2.2.3. Inlet Air Temperature and Hot-Air Flow Rate

We determined microencapsulation efficiency and powder yield at inlet air temperatures of 100 °C, 120 °C, 140 °C, 160 °C, and 180 °C and hot-air flow rates of 0.3, 0.4, 0.5, and 0.6 m^3^/min. We selected the optimal inlet air temperature and hot-air flow rate.

### 2.3. Determination of Indicators of Microcapsules Containing Insect Wax–Derived Policosanol

#### 2.3.1. Microencapsulation Efficiency and Powder Yield

We extracted the policosanol from the surface of microcapsules as follows [26]: The microcapsules (1.25 g) were added into 20 mL chloroform, shaken for 10 min, and filtered. We repeated this process three times and combined the filtrates, which were concentrated to 5 mL. We determined the content of policosanol using gas chromatography under the following conditions: column temperature 290 °C, inlet temperature 290 °C, flame ionization detector temperature 300 °C, and split ratio 1:30.

We extracted the policosanol from the surface and core of microcapsules as follows: The microcapsules (0.50 g) were added into 10 mL HCl solution (0.1 mol/L) and ultrasonicated for 15 min to release the inner substances. The mixture was added to 20 mL chloroform, shaken for 10 min, and filtered. We collected the chloroform layer and repeated this process three times. The chloroform layers were combined and concentrated to 5 mL. We determined the content of policosanol using gas chromatography.

We calculated microencapsulation efficiency and powder yield using the following equations [26,27,28]:(1)$$Microencapsulation\;efficiency\left(\%\right)=(1-\frac{m_{s}}{m_{t}})\times 100$$
(2)$$Powder\;yield\left(\%\right)=\frac{m_{3}}{m_{1}+m_{2}}\times 100$$
where *m_s_* is the weight of policosanol on the surface of microcapsules/g, *m_t_* is the weight of policosanol on the surface and in the core of microcapsules/g, *m*_1_ is the weight of policosanol added before spray drying/g, *m*_2_ is the weight of wall materials and emulsifier/g, and *m*_3_ is the weight of microcapsules obtained after spray drying/g.

#### 2.3.2. Water Content and Water Dispersion

We determined the water content of microcapsules using a rapid moisture meter (HX204, Mettler Toledo, Zurich, Switzerland). Microcapsules (5.00 g) were added to 50 mL of water (30 °C) and gently stirred. We recorded the time required for the microcapsules to be completely dispersed in water. Microcapsules (3.00 g) were stirred and dispersed in 40 mL of water (30 °C) and centrifuged at 4000 rpm for 15 min. After we collected the upper layer suspension, we added water (40 mL) in several portions and stirred. The mixture was centrifuged at 4000 rpm for 15 min. The upper layer suspension was dried at 105 °C to a constant weight. We calculated water dispersion (*WD*) using the following equation [29]:(3)$$WD\left(\%\right)=(1-\frac{m_{1}}{(1-WC)\times m})\times 100$$
where *m* is the weight of microcapsules (g), *m*_1_ is the weight of the dried upper layer suspension (g), and *WC* is the water content of microcapsules (%).

#### 2.3.3. Bulk Density and Surface Morphology

The bulk density of microcapsules was measured according to the method described by Rodríguez-Cortina [30] with slight modification. 2.00 g microcapsules were transferred to a 10 mL graduated cylinder. The bulk density was calculated according to the mass and volume of microcapsules. The morphology of microcapsules was observed using a scanning electron microscopy (Hitachi TM3000, Tokyo, Japan). The samples were coated with gold nanoparticles using a small magnetron sputtering instrument (JS-1600, Beijing Hetong Chuangye Technology Co., Ltd., Beijing, China). The images were taken at an acceleration voltage of 15 kV.

#### 2.3.4. Residence Time of Microcapsules in Simulated Stomach Fluid

Microcapsules (4.00 g) were added into 19 mL of HCl solution (0.1 mol/L) and slowly stirred at 37 °C. We observed the microcapsules under a microscope every 10 min until the microcapsules completely disappeared. We determined the residence time of microcapsules in simulated stomach fluid [31].

### 2.4. Preparation of Effervescent Tablets Containing Insect Wax–Derived Policosanol

We used the previously prepared microcapsules in the preparation of the effervescent tablets containing insect wax–derived policosanol. The mixture containing microcapsules (2.50 g), vitamin C (0.01 g), sucrose (0.40 g), citric acid (6.00 g), lemon yellow pigment (0.30 g), and maltodextrin (2.00 g) was passed through an 80-mesh sieve. We added the mixture to 4 mL of ethanol (50%, *v/v*), which was thoroughly mixed and passed through a 20-mesh sieve for granulation. The granules were dried at 55 °C to obtain granule A. 8.00 g of sodium bicarbonate was added to 4 mL of ethanol (50%, *v/v*), which was thoroughly mixed and passed through a 20-mesh sieve for granulation. The granules were dried at 50 °C to obtain granule B. Then, 3.00 g granule A, 2.00 g granule B, and 0.80 g polyethylene glycol 6000 were ground and well mixed to prepare the effervescent tablets (0.50 g each tablet) using a tablet compression machine (769YP-15A, Tianjin Keqi High tech Co., Ltd., Tianjin, China).

### 2.5. Determination of Quality Indicators of Effervescent Tablets Containing Insect Wax–Derived Policosanol

#### 2.5.1. pH

We dissolved one effervescent tablet in 100 mL water and measured the pH with a pH meter (CPC-505, Smarttester, Germany) at 25 °C.

#### 2.5.2. Friability and Hardness

According to the General Chapter 0923, Tablet Friability Test Method, in the *Chinese Pharmacopoeia* (Part IV) [32], we determined the friability and hardness of the effervescent tablets using a tablet friability and hardness tester (CJY-2C, Shanghai Huanghai Pharmaceutical Testing Instrument Co., Ltd., Shanghai, China).

#### 2.5.3. Disintegration Time

According to the General Chapter 0921, Disintegration Time Test Method, in the *Chinese Pharmacopoeia* (Part IV) [32], we placed an effervescent tablet into 200 mL of water at 20.0 ± 5.0 °C. The tablet was allowed to dissolve completely until no gas bubbles were escaping and no aggregated particles remained in the water. We recorded the time as the disintegration time and repeated the measurement six times.

#### 2.5.4. Bubble Volume

We added one effervescent tablet (0.50 g each tablet) to 50 mL of water at 20.0 ± 5.0 °C. The produced gas was immediately introduced into an inverted 10-mL measuring cylinder filled with water. Once no more gas was generated, we recorded the volume of gas in the measuring cylinder as the bubble volume.

## 3. Results and Discussion

### 3.1. Optimization of Conditions for Preparation of Microcapsules Containing Insect Wax–Derived Policosanol

#### 3.1.1. Effects of Wall Material Ratio and Core-to-Wall Material Ratio on Microencapsulation

Wall materials play a pivotal role in determining the characteristics of microcapsules. The water solubility, emulsification capability, drying property, film-forming property, and viscosity of wall materials significantly influence microencapsulation. We conducted a preliminary experiment to screen for wall materials. We selected maltodextrin, starch sodium octenyl succinate, and HPβCD for the microencapsulation of insect wax–derived policosanol. The ratio of these materials demonstrated a notable impact on microencapsulation efficiency (Figure 2A). The addition of starch sodium octenyl succinate positively affected microencapsulation efficiency. This could be attributed to its hydrophobic and hydrophilic nature, leading to excellent emulsification stability [33]. This resulted in the formation of a continuous and robust film at the oil-water interface. During spray drying, starch sodium octenyl succinate rapidly coagulated and solidified on the surface of core materials, showing remarkable synergistic effects with other emulsifiers, thereby achieving high microencapsulation efficiency. Starch sodium octenyl succinate is commonly used for the encapsulation of water-insoluble substances [34,35]. A further increase in starch sodium octenyl succinate, however, reduced microencapsulation efficiency, probably because of the increase in emulsion viscosity that hindered the spray-drying process. Moreover, HPβCD is water-soluble and possesses a hydrophobic cavity for encapsulating fat-soluble substances, which is a suitable wall material for encapsulating water-insoluble substances. As the proportion of starch sodium octenyl succinate increased, the proportion of HPβCD decreased, which resulted in a decline in microencapsulation efficiency. The optimum wall material ratio, which resulted in the highest microencapsulation efficiency and powder yield, was achieved at a mass ratio of 1:1:1 for maltodextrin, starch sodium octenyl succinate, and HPβCD. We selected this ratio as optimal for microencapsulation.

As shown in Figure 2B, the core-to-wall material ratio, which ranged from 1:3 to 1:12, had a minimal impact on powder yield but significantly influenced microencapsulation efficiency. At core-to-wall material ratios of 1:3 and 1:5, microencapsulation efficiency was below 60%. This could be attributed to a relatively low proportion of wall material available for forming the capsule wall, which resulted in reduced microencapsulation efficiency. Microencapsulation efficiency increased as the proportion of wall material increased. At a core-to-wall material ratio of 1:10, microencapsulation efficiency reached 90%. Further increasing the proportion of wall material did not significantly affect microencapsulation efficiency and powder yield. Therefore, a core-to-wall material ratio of 1:10 was appropriate for microencapsulation.

#### 3.1.2. Effects of Solid Content and Feed Temperature on Microencapsulation

Solid content plays a crucial role in microencapsulation. As shown in Figure 3A, when the solid content was below 15%, the microencapsulation efficiency increased with solid content. Beyond 15% solid content, however, the microencapsulation efficiency started to decrease. This decrease could be attributed to the fact that a larger amount of water had to be removed during the spray-drying process at low solid content, which hindered the formation of the capsule wall [26]. Appropriately increasing the solid content aided the formation of a robust capsule wall and enhanced its densification during spray drying, which improved microencapsulation efficiency. Nonetheless, excessively high solid content increased the viscosity of feed emulsion, which would produce large particles during drying. The large particles were dried more slowly than the small particles. As a consequence, both encapsulation efficiency and powder yield decreased [36,37]. This result was similar to the findings of Zhang et al. [26]. Therefore, we selected a solid content of 15%.

Feed temperature plays a significant role in feed emulsification and fluidity, consequently affecting the drying rate and microencapsulation efficiency. As shown in Figure 3B, powder yield showed little sensitivity to feed temperature, whereas microencapsulation efficiency was improved at higher feed temperatures. Feed temperatures higher than 60 °C, however, did not significantly affect microencapsulation efficiency. To reduce the cost of production, we selected 60 °C as feed temperature. This temperature provided a suitable emulsion viscosity and allowed for drying microcapsules at the wet-bulb temperature.

#### 3.1.3. Effects of Inlet Air Temperature and Hot-Air Flow Rate on Microencapsulation

The research of Gutiérrez-López et al. [38] showed that the stability of the emulsions changed after their passage through the pump and nozzle during the spray drying process, which in turn affected the microencapsulation efficiency and yield. The drying rate, water content, particle structure, and stability of the heat-sensitive substances in microcapsules were affected by the inlet air temperature and hot-air flow rate. As shown in Figure 4A, when the temperature of inlet air was below 140 °C, the microencapsulation efficiency increased with the increase in temperature, and the efficiency was the highest at 140 °C. A low inlet air temperature led to a slow drying rate, which resulted in insufficient density and strength of the wall and thus poor encapsulation of the core material. The same results were obtained in the study of Maury et al. [37] and Temiz et al. [36]. A higher inlet air temperature accelerated water evaporation and wall formation, which enhanced the encapsulation of the core material [39]. When the inlet air temperature exceeded 140 °C, the microencapsulation efficiency decreased notably. Excessively high inlet air temperature caused rapid volatilization of the solvent on the droplet surface, thus forming a hard shell with imbalanced inside and outside temperatures. The rapid evaporation of residual solvent inside the wall led to a wall rupture or depression, which negatively affected the encapsulation efficiency. Considering both microencapsulation efficiency and powder yield, the inlet air temperature was selected to be 140 °C.

As shown in Figure 4B, the hot-air flow rate ranged from 0.3 to 0.5 m^3^/min, and both microencapsulation efficiency and powder yield increased with the increase of the hot-air flow rate. A higher hot-air flow rate improved the drying rate and reduced the heating time of microcapsules, which facilitated the encapsulation of the core material. However, when the hot-air flow rate exceeded 0.5 m^3^/min, both microencapsulation efficiency and powder yield started to decline. An excessively high hot-air flow rate led to overheating of the product, which resulted in damage to the active ingredients and unfavorable conditions for encapsulating the core material. Furthermore, a very high hot-air flow rate increased the speed of flow at the tower outlet, and the microcapsules in the separation chamber could be easily carried out, which resulted in a reduction of powder yield. Consequently, we selected the hot-air flow rate of 0.5 m^3^/min for this experiment.

### 3.2. Quality Indicators of Microcapsules Containing Insect Wax–Derived Policosanol

We prepared microcapsules containing insect wax–derived policosanol, which resulted in a white powder with fine particles and excellent flowability (Figure 5A). When viewed under a microscope, the microcapsules exhibited a regular spherical shape (Figure 5B–D). These microcapsules had low water content (1.82%) and thus resisted agglomeration and fungal infection. Moreover, they demonstrated rapid dispersion in water (within 143.5 s) and high water dispersion (96.5%). These properties facilitated the full utilization of the biological functions of insect wax–derived policosanol and ultimately enhanced its bioavailability. The residence time of microcapsules in the simulated stomach fluid was about 60.0 min, which indicated that the policosanol could be released to exert its biological functions upon entering the stomach (Table 1). In addition, the loading capacity of policosanol in the microcapsules was 2.63% (*w/w*), which was much higher than that of microcapsules previously prepared in our laboratory (0.22%) [19].

### 3.3. Effervescent Tablets Containing Insect Wax–Derived Policosanol

An effervescent tablet is a form of solid drink that contains effervescent disintegrant. When placed in water, it releases a significant amount of carbon dioxide, causing rapid disintegration of the tablet. The properties of individual ingredients are very important for the physics and dissolution properties of effervescent tablets [40]. In this study, we prepared yellow effervescent tablets containing insect wax–derived policosanol using the previously obtained microcapsules (Figure 6A). We measured the quality indicators of the tablets, and the results are shown in Table 2. Upon contact with water, the effervescent tablets disintegrated rapidly within 99.5 s (Figure 6B,C), which met the requirement for disintegration time (<5 min) set by the *Chinese Pharmacopoeia* for effervescent tablets [32]. Saifullah et al. prepared four natural fruit powder effervescent tablets, which all took more than 6 min to dissolve in water [40]. It can be seen that the effervescent tablets prepared in this study can be quickly dispersed in water, which saves more time when consumed by consumers. Saifullah et al. reported that the particle size affected the dissolution rate of effervescent tablets [40]. Barbosa-Canovas et al. reported that the presence of free fat on the particle surface reduced the wettability of the powder, which in turn affected the solubility of the effervescent tablets [41]. Therefore, the powder particles and chemical compositions both can affect the dissolution or dispersion of effervescent tablets. As shown in Table 2, each 0.50 g of effervescent tablet generated 8.70 mL of CO_2_ when exposed to water, giving the effervescent tablet solution a fizzy taste in the mouth. The theoretical volume of CO_2_ was calculated to be 46.00 mL according to the following reaction equation between citric acid and sodium bicarbonate:(4)$$3\;NaHCO_{3}+C_{6}H_{8}O_{7}=C_{6}H_{5}O_{7}{Na}_{3}+3\;H_{2}O+3\;CO_{2}$$

The actual volume of CO_2_ measured was less than the theoretical one, possibly because the reaction was incomplete or some of the CO_2_ was dissolved in the water. The pH value was close to neutral, which indicated that the excipient was suitable for the preparation of effervescent tablets. Additionally, the friability of the tablets was 0.32%, which satisfied the requirement for tablets (≤1%) described in the *Chinese Pharmacopoeia* [32]. This also provided the basis for the transportation and portability of the prepared effervescent tablets.

Insect wax is a mixture of lipids derived from *Ericerus pela* (Chavannes) and has important physiological activities. As the primary active component of insect wax, policosanol has the effects of reducing fatigue and enhancing physical strength and endurance. Although policosanol is an excellent antifatigue, its limited solubility and dispersion in water have restricted its application [19]. To address this problem, we developed microcapsules and effervescent tablets containing insect wax–derived policosanol and optimized their preparation conditions. The microcapsules showed high water dispersion, allowing it to be consumed directly or dissolved in water before ingestion. Moreover, the microcapsules containing policosanol can be used in the research and development of pharmaceuticals, cosmetics, and healthcare products, expanding the potential application of insect wax–derived policosanol and the utilization of its biological functions. By using the microcapsules, we formulated effervescent tablets that, when placed in water, created an effervescent sports drink containing insect wax–derived policosanol. The taste of the effervescent drink resembled that of carbonated drinks. The loading capacity of policosanol in microcapsules was 2.63% (*w*/*w*) and each 0.50 g effervescent tablet contained 1.64 mg policosanol. According to a study by NOF Corporation, oral intake of 0.20–0.80 mg policosanol per person per day can relieve fatigue and improve endurance and physical strength [19]. Therefore, one tablet per person per day should be effective for relieving fatigue. The microcapsules and effervescent tablets prepared in this study offered several advantages, including convenient storage, high portability, easy ingestion, high water dispersion, and high bioavailability. The taste was modified by mixing the microcapsules with other excipients, overcoming the discomfort associated with the consumption of insect wax and increasing its acceptance among consumers. These findings offer a novel approach to the processing, formulation, and promotion of insect lipids.

## 4. Conclusions

To address the negligible water solubility, low absorption rate, and limited psychological acceptance of *Ericerus pela* (Chavannes) lipids consumption, we developed two new products, microcapsules and effervescent tablets containing insect wax–derived policosanol which was the key active component of *Ericerus pela* (Chavannes) functional lipids. Both products showed excellent water dispersion, effectively addressing the water insolubility of insect wax and its active component policosanol. Moreover, they offered promising solutions for the secondary processing and consumption of insect lipids. We optimized the conditions of preparing microcapsules by spray drying and achieved 92.09% microencapsulation efficiency and 61.67% powder yield under the following conditions: wall materials maltodextrin, starch sodium octenyl succinate, and HPβCD at a ratio of 1:1:1, core-to-wall material ratio of 1:10, 15% solid content, feed temperature of 60 °C, inlet air temperature of 140 °C, and hot-air flow rate of 0.5 m^3^/min. The microcapsules containing policosanol exhibited a spherical shape with low water content. In the simulated stomach fluid, policosanol was released quickly to exert its biological functions. The policosanol-containing effervescent tablets showed rapid disintegration, low friability, high bubble volume, and a pH close to neutral after dissolution. The tablets demonstrated excellent effervescent characteristics, which aligned with the standards described in *Chinese Pharmacopoeia*. The tablets hold considerable potential for a wide range of applications. In summary, this work proposed a new strategy for the functional lipids consumption of *Ericerus pela* (Chavannes).

## Figures and Tables

**Figure 1 foods-12-03567-f001:**
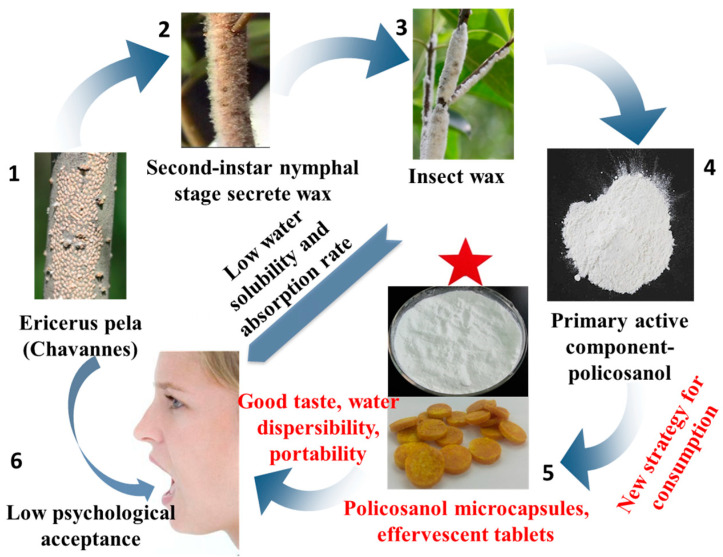
A new strategy for the consumption of lipids from *Ericerus pela* (Chavannes).

**Figure 2 foods-12-03567-f002:**
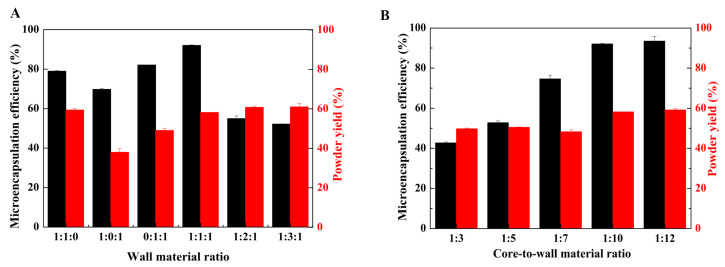
Effects of (**A**) wall material ratio and (**B**) core-to-wall material ratio on microencapsulation efficiency and powder yield.

**Figure 3 foods-12-03567-f003:**
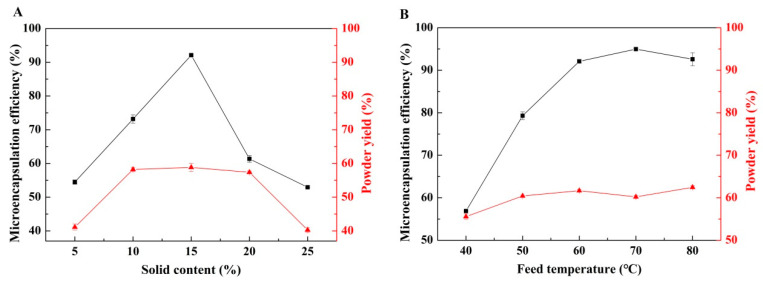
Effects of (**A**) solid content and (**B**) feed temperature on microencapsulation efficiency and powder yield.

**Figure 4 foods-12-03567-f004:**
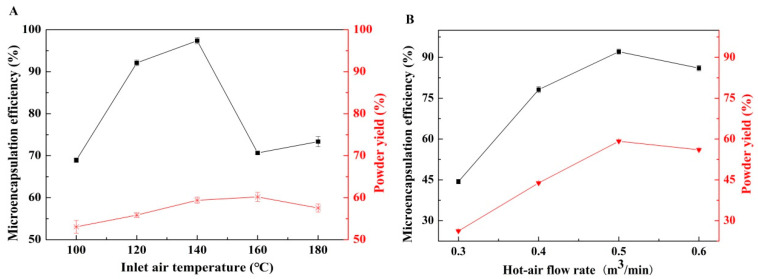
Effects of (**A**) inlet air temperature and (**B**) hot-air flow rate on microencapsulation efficiency and powder yield.

**Figure 5 foods-12-03567-f005:**
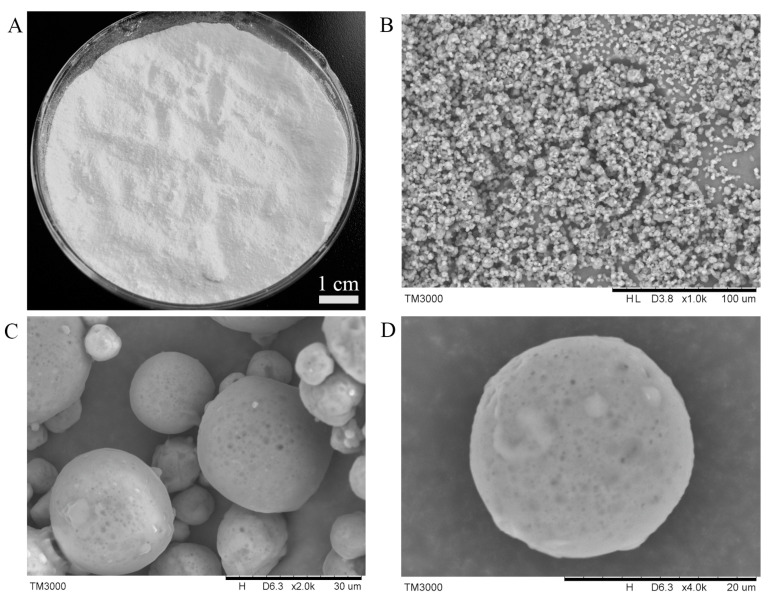
(**A**) Photos and (**B**–**D**) scanning electron microscope images of microcapsules containing insect wax–derived policosanol with magnifications of 1000× (**B**), 2000× (**C**), and 4000× (**D**).

**Figure 6 foods-12-03567-f006:**
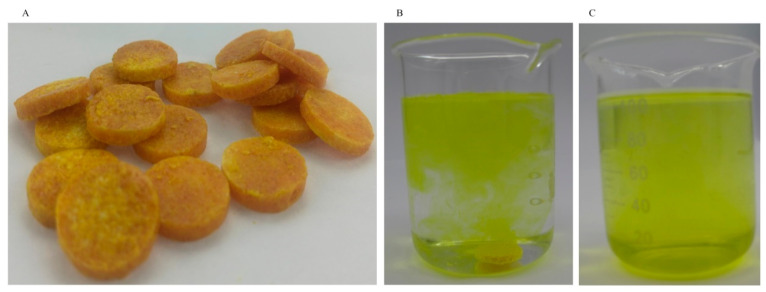
Effervescent tablets containing (**A**) insect wax–derived policosanol, (**B**) disintegration of effervescent tablets in water, and (**C**) complete disintegration of effervescent tablets in water.

**Table 1 foods-12-03567-t001:** Quality assessment of microcapsules containing insect wax–derived policosanol.

Property	Value
Water content (%)	1.82 ± 0.44
Water dispersion time (s)	143.5 ± 5.1
Water dispersion (%)	96.5 ± 0.9
Bulk density (g/mL)	0.26 ± 0.01
Residence time in simulated stomach fluid (min)	60.0 ± 2.6

**Table 2 foods-12-03567-t002:** Quality assessment of effervescent tablets containing insect wax–derived policosanol.

Property	Value
pH	6.66 ± 0.14
Friability (%)	0.32 ± 0.01
Hardness (N)	30.52 ± 2.31
Disintegration time (s)	99.50 ± 9.9
Bubble volume (mL)	8.70 ± 1.5

## Data Availability

The data used to support the findings of this study can be made available by the corresponding author upon request.

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
