# Peer review of "A New Strategy for Consumption of Functional Lipids from *Ericerus pela* (Chavannes): Study on Microcapsules and Effervescent Tablets Containing Insect Wax–Derived Policosanol"

_foods, 2023, doi:10.3390/foods12193567_

Round 1

Reviewer 1 Report

The manuscript addresses the development of a new way to consume the active agent (policosanol) of an insect wax. This strategy involves the preparation of a microencapuslated powder and efervescent tablets and their characterization. However, the work lacks  information and techniques that can strengthen the study. I recommend the manuscript after a major revision would be done. 

The comments and suggestions are the following:

-In the introduction section, Fig. 1 is not clear. Arrows of the image are not easily readable. Please use numbers or letters to guide the reading.

-In section 2.2.1: core-to-wall material ratios are not clear how were they determined? Why they had m/m units? Please explain.

-Sections 2.3.3 and 2.5.1: please add more details about the evaluation of Bulk density and surface morphology. Provide equipment brands and work conditions.

-Fig 2. It seems that Figs. 2A and 2B are the same graph.

-Section 3.1.1. the core-to-wall material ratio discussion needs a deep revision because results did not correspond to the Fig. 2B. 

-Section 3.1.2. Some references to support the results must be added.

-Section 3.1.3. The discussion must be improved with comparisons with previous reports in literature. The following article is suggested to complete this section: https://doi.org/10.1016/j.jfoodeng.2022.111056

-Section 3.2: the spherical shape of the powder particles is not easily observable in Fig. 5B. Please add one (or more) micrographs which support this affirmation. It can be images having higher magnifications. Also, Fig. 5B caption did not show the magnification (how many X's?)

-Section 3.2: The "uniformity" of particle sizes was not proved. Please add data regarding particle size.

-Fig. 5A: a scale bar is missing.

-Section 3.3. Discussion is poor, please add more contrasting literature.

Author Response

Please see the attachment。

Reviewer 2 Report

Liu et al. manuscript contains much interesting information and a new approximation on the preparation of effervescent microencapsules. Unfortunately, the submitted version is inappropriate for publication as the described isolation and microencapsulation procedures and effervescent table preparation are not reproducible.

- In the 2.1 section, the authors mention that they obtained policosanol from insect wax via a reduction reaction, but the reduction method is missing. If the reduction is already published, they should provide a short but informative about the procedure used. But, if they have not published the reduction method, a method description is mandatory.

- They used 2-hydroxypropyl-β-cyclodextrin. Firstly, the correct name is (2-hydroxy)propyl-β-cyclodextrin (HPβCD). But, as at least three different HPβCDs are on the market (DS ~3.5, ~4.5, and ~6.3, a Chinese producer probably also sells a product of an even higher DS, and the DS is the Degree of Substitution, according to the pharmacopeias, the number of substituent/CD ring). This information is inevitable as many molecules show significant DS-dependent complexation affinity to HPβCDs.

- Some parameters of used soybean oil and lecithin are needed, as the quality of these products may vary from producer to producer, and finding an equivalent of the used materials on the other end of the world can be challenging.

- The unit r/min is unknown. Does this expression cover the rpm? If yes, please provide the conventional name of the unit.

- In subchapter 2.2, weights and volumes are missing.

- In point 2.2.1, the mass ratios are unclear. While in the second line, the ratios contain three materials, only two ones are identified (maltodextrin, starch sodium octenyl succinate, and HPβCD - by the way, the full name of HPβCD has a new version in that line!).

- In section 2.4, weights are also missing. The amount of NaHCO3 is also inaccurate as the base of the used percent is unknown. On the other hand, the 3.98% of polyethylene glycol 6000 is strange, as the referee is sure that the difference between 4% and 3.98% is negligible.

- The volume of the generated CO2 has no meaning because the weight and composition of the effervescent tablet are unknown. In the case of the known amounts of CO2 generators, the theoretical gas volume can also be calculated and compared to the measured value. It is not a serious drawback but could be informative.

The language seems correct, but a minor spell check is necessary.

Round 2

Reviewer 2 Report

The manuscript is suitable for publication as the authors answered the referee's concerns correctly.